# High-versus conventional-volume pericapsular nerve group (PENG) block for total hip arthroplasty: A randomized, controlled trial

Qian-song Wang[1], Shi-ming Qin[1], Yang Zhao[2], Chong-mei Gao[1], Xia Yuan[1], Zhao-hui Chen [1]*

**1** Department of Anesthesiology, Third Affiliated Hospital of Chongqing Medical University, Chongqing, China, **2** Department of Anesthesiology, Affiliated Hospital of North Sichuan Medical College, Nanchong, Sichuan, China

\* 650688@cqmu.edu.cn

## Abstract

### Background

Currently, the optimal volume of pericapsular nerve group (PENG) block for analgesia after total hip arthroplasty (THA) has not been clarified. In this trial, we investigated whether a high-volume PENG block has a superior analgesic effect than a conventional PENG block for primary THA.

### Methods

Forty patients receiving primary THA under spinal anesthesia were enrolled and randomly divided into the high-volume PENG group (40 mL of 0.375% ropivacaine) and the conventional-volume PENG group (20 mL of 0.375% ropivacaine). Dexamethasone (5 mg) was added to the local anesthetic in both groups. A blinded researcher performed pain scores and lower limb sensory and motor block assessments at 3, 6, and 24 h after surgery. The time of first walking, time of first opioid consumption, total opioid consumption within 48 h, nerve block, opioid-related complications, and length of hospital stay were recorded. The primary outcome was the dynamic pain scores at 6 h post-surgery.

### Results

Thirty-seven patients were included in the final analysis, among whom 18 were in the high-volume group and 19 in the conventional-volume group. There were no significant differences between the two groups in dynamic pain scores at 6h after surgery [median and interquartile range (p25, p75) 5(4,6) vs 4 (4,5)] and other secondary outcomes.

**Data availability statement:** All relevant data are within the paper and its Supporting information files.

**Funding:** The author(s) received no specific funding for this work.

**Competing interests:** All authors declare no conflicts of interest.

## Conclusions

The high-volume PENG block is not superior to conventional-volume PENG block in improving post-operative analgesia in patients undergoing primary THA and does not increase the risk of quadriceps weakness.

## Trial registration

Chinese Clinical Trial Registry (ChiCTR, https://www.chictr.org.cn). Clinical trial registration number: ChiCTR2300077281.

## Introduction

The rate of total hip arthroplasty (THA) has been on the rise worldwide [1–3]. Patients undergoing THA often experience moderate or even severe post-operative pain, limiting their recovery [4–7]. The lumbar plexus block, fascia iliaca block, and femoral nerve block are the most commonly used nerve blocks for multimodal analgesia in patients receiving THA. However, such regional blocks can potentially weaken the quadriceps and hinder post-operative functional exercise.

The pericapsular nerve group (PENG) block, first described in 2018 by Girón-Arango *et al.*, can reduce static and dynamic pain scores and preserve motor function of the affected limb in patients with hip fractures. Mechanistically, the PENG block selectively blocks the sensory branches of the anterior hip capsule of the femoral nerve and the accessory obturator nerve between the anterior inferior iliac spine and the iliopubic eminence [8]. Moreover, the PENG block can decrease post-operative pain score in patients after the primary THA [9], improve the quality of recovery, and accelerate patient rehabilitation [10].

Most PENG blocks are performed with a local anesthetic volume of 20 mL, with volumes ≥30 mL considered high. Although conventional-volume PENG block can reduce post-operative pain score and accelerate rehabilitation recovery, the dynamic score of conventional-volume PENG block may reach 6 at 6 h after THA [11], closely matching severe pain, which can potentially limit patient participation in early exercise and rehabilitation training. This indicates that the PENG block requires further optimization. The joint branches of the obturator nerve that innervate the hip joint are located near the inferomedial acetabulum, which is quite distant from the bone surface between the anterior inferior iliac spine and the iliopubic eminence. This anatomical limitation poses challenges in effectively blocking the articular branches of the obturator nerve with the conventional-volume PENG block. Balococo, A.L [12] confirmed that 20 ml of local anesthetic and contrast agent failed to spread to the obturator nerve during PENG block in patients undergoing hip surgery. However, Ahiskalioglu et al. demonstrated that high-volume PENG block can effectively block obturator nerve [13]. Data from other studies have indicated that conventional-volume PENG block may lead to quadriceps myasthenia due to the spread of local anesthetics to the femoral nerve in some cases [14,15]. Therefore, we speculate that while the analgesic efficacy of a high-volume PENG block may be enhanced, it may also increase the risk of lower

limb motor blockade in more patients. Notably, the high-volume PENG block may still be applied in clinical treatment if it enhances the analgesic effect without significantly increasing the incidence of quadriceps weakness. To date, studies comparing the effects of high-versus conventional-volume PENG blocks on post-operative pain and rehabilitation in patients undergoing THA are scarce. This randomized controlled trial aimed to compare the effects of high-volume versus conventional-volume PENG blocks on postoperative pain and rehabilitation outcomes in patients undergoing THA. Additionally, the study sought to explore the optimal block volume for this surgical procedure. We hypothesized that the dynamic pain score at 6 h after THA might be lower in patients receiving the high-volume PENG block compared to those receiving the conventional-volume PENG block. Secondary outcomes included dynamic pain scores at 3 h and 24 h post-surgery, static pain scores, sensorimotor block assessments at 3 h, 6 h and 24 h, first walking time, opioid consumption within 48h after surgery, length of hospital stay, and anesthesia-related complications.

## Materials and methods

### Study design and patients

This was a randomized, controlled trial was conducted at the Third Affiliated Hospital of Chongqing Medical University from August to November, 2024. This randomized trial was registered at the Chinese Clinical Trial Registry (https://www.chictr.org.cn/; ChiCTR2300077281; November 3, 2023) before patient recruitment. The study was approved by the ethics committee of the third affiliated hospital of Chongqing Medical University and was conducted in accordance with the principles of the Declaration of Helsinki. The first patient was enrolled on August 29, 2024, and the last patient was followed up on November 30, 2024.The study protocol followed Consolidated Standards of Reporting Trials (CONSORT) guidelines.

All enrolled patients provided written informed consent to participate. Patients receiving primary THA at the third affiliated hospital of Chongqing Medical University were enrolled using the following criteria: (1) Patients aged ≥18 years old; (2) those classified as American Society of Anesthesiologists physical status (ASA) grades 1–3; (3) those with a body weight of 45–90 kg; (4) those receiving primary unilateral THA due to end-stage hip diseases or hip fracture, a posterior lateral approach; (5) those with no contraindications to regional or intravertebral anesthesia; and (6) those who could reliably report symptoms to the researchers. Exclusion criteria: (1) Patient refusal; (2) those with a history of local anesthetic allergy; (3) those with infection near the puncture site: (4) those with dementia or cognitive impairment; (5) those with moderate or severe anemia; (6) pregnant patients; (7) those with chronic pain; and (8) those with long-term intake of painkillers such as opioid.

### Randomization and blinding

A nurse anesthetist who was not involved in the study randomly divided patients into two groups with approximately equal sample sizes using a random number table in a 1:1 ratio: the high-volume group (even numbers) and the conventional-volume group (odd numbers). The grouping data were sealed in sequentially numbered opaque envelopes, which were opened on the day of surgery by researchers who performed the nerve block. All nerve blocks were performed by a trained anesthesiologist skilled in PENG blocks. Following the procedure, another anesthesiologist from the acute pain service team, who was not involved in the nerve block or intraoperative management and was blinded to the patients' group assignments, assessed the patients post-surgery. This assessment included evaluating post-operative pain scores as well as sensory and motor blocks, and collecting relevant data. Unblinding occurred only after the data collection process was completed for all patients.

### Performance of PENG blocks

The ultrasound-guided PENG block was performed preoperatively, with routine monitoring of blood pressure, oxygen saturation, and electrocardiogram; the patient was placed in the supine position. The ultrasound convex array probe (2–5 Hz)

(HITACHI ALOKA ARIETTA, Fujifilm Medical, Tokyo, Japan) was positioned on transverse orientation, medial and caudal to the anterosuperior iliac spine to identify the anteroinferior iliac spine, the iliopubic eminence and the psoas tendon. Using an in-plane technique and a lateral to medial orientation, the block needle (B. Braun Melsungen AG, Melsungen, Germany) was advanced until its tip was located on the periosteum on the dorsal side of the iliopsoas tendon. After negative aspiration, the high-volume group was injected with 40 mL of 0.375% ropivacaine, and the conventional-volume group received 20 mL of 0.375% ropivacaine, resulting in a local anesthetic between the periosteum of the iliopsoas tendon and iliopubic eminence (Fig 1). Dexamethasone (5 mg) was mixed with the local anesthetic and administered simultaneously in each group.

## Anesthetization protocol, surgery, and analgesia regimen

After completion of the nerve block, a spinal anesthesia puncture was performed immediately in the lateral position at the L3-4 interspace. Subsequently, 10 mg of bupivacaine (volume = 2 mL) was injected, and 1.5 μg/kg/min dexmedetomidine was continuously injected for calm during the operation. Both groups were given a patient-controlled analgesia electronic pump (no continuous background dose, 0.16 mg of hydromorphone bolus; lockout interval = 10 min) at the end of the operation and were trained for breakthrough pain(VAS pain scores≥4) in the ward. Subjects were also given a regulation injection of acetaminophen (0.5 g every 6 h) in the ward.

## Outcomes

The primary outcomes were the dynamic pain scores (with hip adduction) reported by patients using a 0–10 VAS (0 indicated no pain and 10 indicated worst pain imaginable) at 6 h post-surgery. Secondary outcomes included static pain scores (at rest) at 3 h, 6 h, and 24 h post-surgery, dynamic pain scores of patients at 3h and 24 h post-surgery, sensory and motor block of lower limbs at 3 h, 6 h, and 24 h post-surgery. Other secondary outcomes included the time of first opioid consumption (time of first analgesia pump press), total opioid consumption within 48 h, opioid-related complications (e.g., nausea, vomiting, dizziness, pruritus, and respiratory depression), time of first walking, length of hospital stay, falls during hospitalization, post-operative infection, local anesthetic poisoning during nerve block, and vascular puncture.

Post-operative sensory block was evaluated in the anterior, lateral and medial aspects of the mid-thigh innervated by the femoral nerve, lateral femoral cutaneous nerve, obturator and femoral nerve, respectively at 3 h, 6 h, and 24 h post-surgery using the Aliste J's method. For each region, the blockade was evaluated on a 3-point scale: 2 = no block,

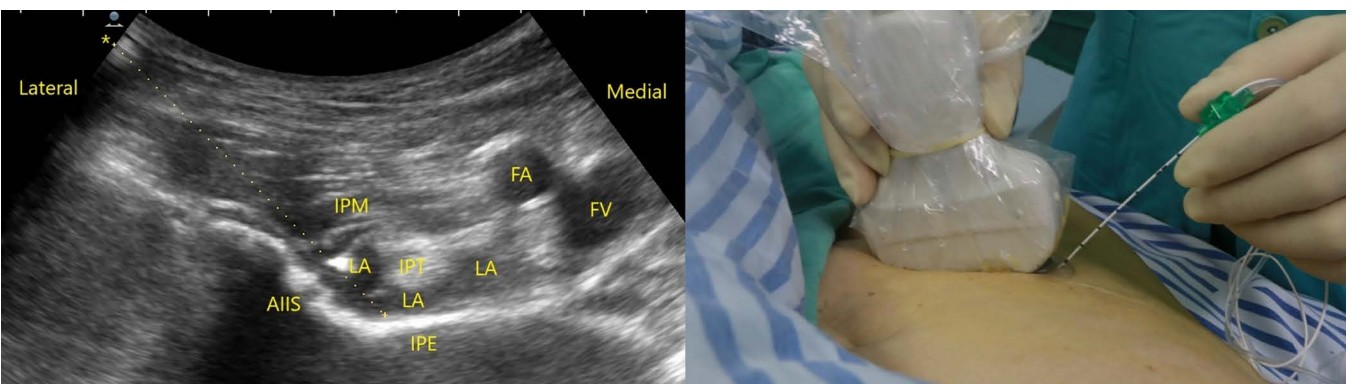

**Fig 1. Ultrasound-guided pericapsular nerve group (PENG) block using an in-plane method.** The needle is advanced to reach the outside of the iliopsoas tendon **(IPT)** and contact the periosteum of iliopubic eminence **(IPE).** Next, a PENG block is achieved by injecting a local anesthetic solution, which can be visualized between the iliopsoas tendon and **IPE. Arrow,** needle pathway**; AIIS,** anterior infer ioriliac spine**; IPM,** iliopsoas muscle**; FA,** femoral artery**; FV,** femoral vein.

1 = analgesia (the patient can feel touch, not cold), 0 = anesthesia (patient cannot feel touch). Post-operative motor block was investigated through knee extension, which was examined while in the supine position with the patient's hip and knee flexed at 45° and 90°, respectively. The patient was asked to extend the knee against gravity and resistance. The extension was graded based on a 3- point scale: 2 = no block (extension against gravity and resistance); 1 = paresis (extension against gravity but not against resistance); 0 = paralysis (no extension possible). First walking was defined as the ability to take at least three steps for the first time post-operatively using a walker. Post-operative consumption of intravenous hydromorphone was converted into intravenous morphine equivalent at the ratio of 1.5: 10.

## Sample size calculation

Sample size calculation was performed using PASS version 11 (Power Analysis and Sample Size Software). The minimum clinically important difference in pain scores was determined based on patients' baseline pain levels. For individuals experiencing moderate pain, a change of 1.3 on the VAS is the threshold for minimum clinically significant pain relief. In contrast, for those with severe baseline pain, a reduction of 1.8 on the VAS is deemed clinically significant [16]. A recent clinical trial reported that the dynamic pain score at 6 h after surgery in patients with a PENG block was 6 [11], which was considered between moderate and severe pain. Therefore, we chose 1.5 as the clinically meaningful minimum dynamic pain score reduction for patients receiving the PENG block. Based on previous data on the use of the PENG block in THA [9,17], the standard deviation (SD) was estimated to be 1.5. To test the difference in pain score of 1.5 (out of 10) and SD of 1.5, a two-tailed, independent samples t-test with an α-error of 0.05 and a β-error of 0.2 was conducted; a minimum of 17 patients were required in each group. Considering the imbalance of grouping and drop-outs, 40 patients were recruited.

## Statistical analyses

SPSS Ver. 20.0 (IBM Corp. Armonk, NY, United States) was used for statistical analyses. The normality of continuous data was assessed using the Shapiro-Wilk test. For normally distributed continuous data, results are presented as mean ± standard deviation (SD) and compared using a two-sample independent t-test. Non-normally distributed continuous variables are expressed as median and interquartile range [M (P25, P75)] and compared using the Mann-Whitney U test. Categorical or graded data were expressed as percentages and compared using the χ² test or Fisher's exact test as appropriate. A two-sided P < 0.05 was considered statistically significant. The log-rank test and Cox regression were used to evaluate time-to-event data.

## Results

Sixty-two patients scheduled for THA were screened for eligibility. Of these, 18 were excluded for not meeting the inclusion criteria, and 4 declined to participate. The remaining 40 patients were randomized to the conventional-volume group (n = 21) and the high-volume group (n = 19). One patient in the high-volume group withdrew from the study due to a failed spinal canal puncture, and 2 patients in the conventional-volume group withdrew from the study due to a change in surgical protocol. The remaining 37 patients were included in the final analysis, with no patients lost to follow-up (Fig 2). Demographic and baseline characteristics, operation time, and the surgical side ratio were comparable between the two groups (Table 1).

The dynamic pain scores of the high- and conventional-volume groups at 6 h after surgery were 5.0(4.0,6.0) and 4.0(4.0,5.0), respectively, indicating that their dynamic pain scores showed no statistically significant differences (Table 2). There were no significant differences between the two groups in dynamic pain scores at 3 and 24 hours, static pain scores at any time point, dermatomal sensory block, or motor block on the surgical side. (Tables 2 and 3).

Four patients (21.1%) in the conventional-volume group and one patient (5.6%) in the high-volume group did not require any opioids for 48 h. There were no significant differences in the proportion of opioid requirements, time of primary

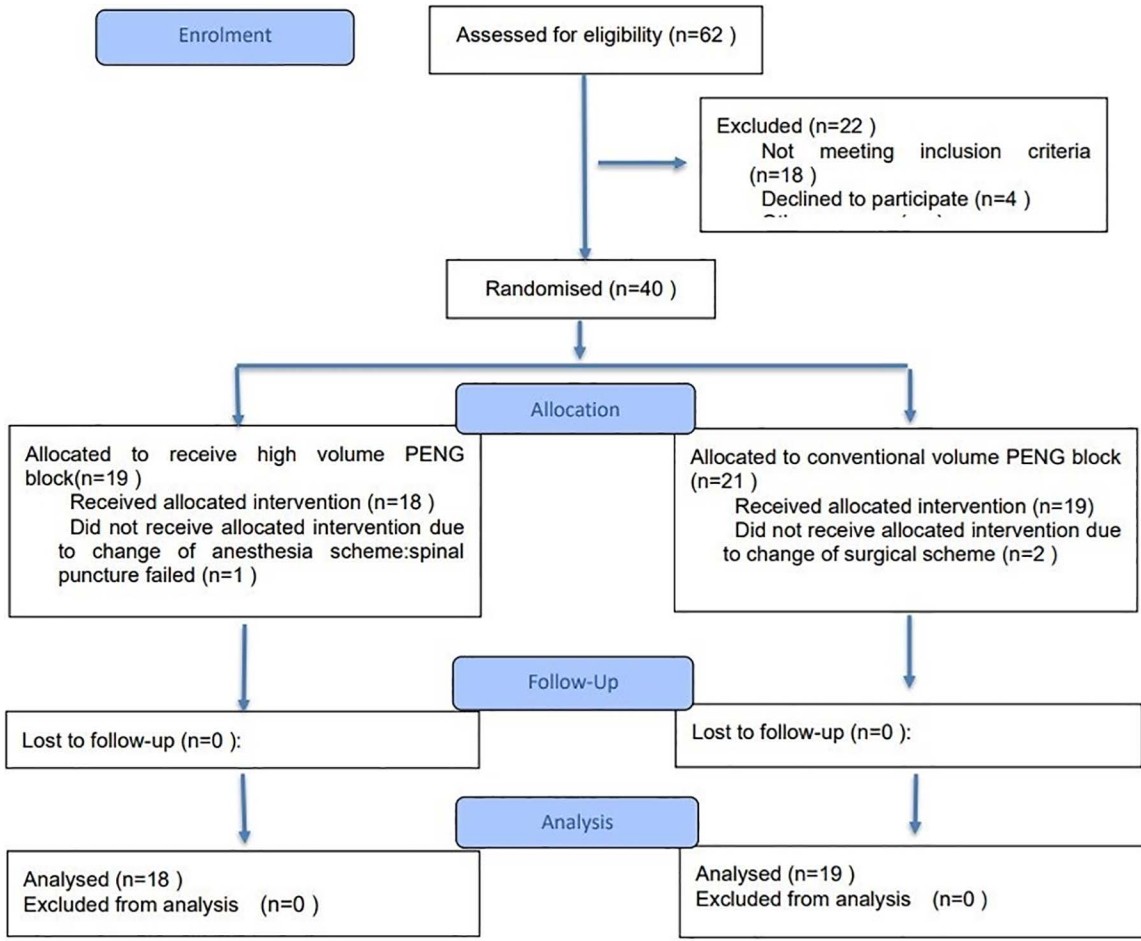

**Fig 2. Flow diagram of study.**

opioid consumption, total opioid consumption within 48 h, time of first walking, length of hospital stay and incidence of post-operative opioid-related complications between the two groups (Table 4). Post-operative respiratory depression, in-hospital falls, wound infections, intravascular injections, and local anesthetic poisoning were observed in neither group during nerve block (Table 4).

## Discussion

In this randomized trial, we compared high-volume versus conventional-volume PENG blocks in patients undergoing primary THA. At 6 h post-operatively, the two groups exhibited no statistically significant differences in dynamic pain scores. Furthermore, dynamic pain scores at other time points and static pain scores throughout the entire postoperative period were statistically comparable between the groups.

Ahiskalioglu *et al.* found that the high-volume PENG block showed the characteristics of a lumbar plexus block [13,18]. Long-acting local anesthetics used in nerve blocks can provide post-operative analgesia for over 10 h. Consequently, it is expected that pain scores within the first 6 h post-operatively would be significantly lower with high-volume PENG blocks compared to conventional-volume PENG blocks. However, this expectation was not supported by the results of the present study. In the research conducted by Ahiskalioglu *et al*, the needle tip was positioned on the medial side of the iliopsoas

**Table 1. Patient characteristics.**

| | Conventional volume group (n=19) | High volume group (n=18) |
|---|---|---|
| Age (years) | 70.0 (56.0,75.0) | 68.5 (55.8,73.3) |
| Gender, n (%) | | |
| Male | 8 (42) | 7 (58) |
| Female | 11 (39) | 11 (61) |
| Weight(kg) | 64.2 (12.7) | 59.6 (8.7) |
| Height (cm) | 159.8 (9.2) | 157.9 (7.5) |
| BMI (kg/m2) | 25.1 (7.1) | 23.9 (3.2) |
| ASA physical status (I/II/III) | 4/7/8 | 4/9/5 |
| Duration of surgery (min) | 85 (72,110) | 88 (75,118) |
| Preoperative use of opioid analgesics (use/no use) | 0/19 | 3/15 |
| Surgical side (left/right) | 12/7 | 7/11 |
| Preoperative pain score | | |
| Static | 1 (0,2) | 1 (0,3) |
| Dynamic | 5 (4,6) | 6 (5,8) |
| Preoperative diagnosis (fracture/no fracture) | 6/13 | 9/9 |

Data are expressed as median (P25,P75), number (proportion) or mean (SD).

ASA, American Society of Anesthesiologists; BMI, body mass index.

**Table 2. Post-operative pain scores.**

| | Conventional volume group (n=19) | High volume group(n=18) | Median difference (95% CI) | P value |
|---|---|---|---|---|
| Post-operative pain; Static VAS | | | | |
| At 3h | 2.0 (0.0,3.0) | 2.0 (1.0,3.0) | 0.0 (−1.0,1.0) | 0.58 |
| At 6h | 2.0 (2.0,4.0) | 3.0 (2.0,4.3) | −1.0 (−1.0,1.0) | 0.33 |
| At 24h | 3.0 (1.0,4.0) | 4.0 (1.8,5.0) | −1.0 (−2.0,1.0) | 0.41 |
| Post-operative pain; dynamic VAS | | | | |
| At 3h | 4.0 (3.0,4.0) | 4.0 (3.0,4.3) | 0.0 (−1.0,1.0) | 0.90 |
| At 6h | 4.0 (4.0,5.0) | 5.0 (4.0,6.0) | 0.0 (−1.0,0.0) | 0.30 |
| At 24h | 5.0 (3.0,7.0) | 6.0 (3.8,6.0) | 0.0 (−2.0,1.0) | 0.64 |

Data are presented as median (P25, P75) and median difference (95% CI).

tendon, allowing the local anesthetic to spread along the psoas major and pubic muscles. This spread effectively blocked the obturator nerve, lateral femoral cutaneous nerve, and femoral nerve. Therefore, the characteristics resembling a lumbar plexus block may be related to suboptimal PENG block..

Elsewhere, Girón-Arango et al. [15,19] inferred that another reason was that the local anesthetic injection was too superficial. In this study, subcutaneous local anesthetic injections were contraindicated. The tip of the needle was required to reach the bone surface of the iliopubic eminence and stick to the lateral side of the iliopsoas tendon before injecting local anesthetic. In the present study, we did not observe evidence suggesting that increasing volume resulted in consistent blockade of peripheral branches of the lumbar plexus.

**Table 3. Post-operative sensory and motor block assessment.**

| | Conventional volume group (n = 19) | | | High volume group(n = 18) | | | *P* value |
|---|---|---|---|---|---|---|---|
| | **No block** | **Analges-ia** | **Anest-hesia** | **No block** | **Analges-ia** | **Anest-hesia** | |
| **Post-operative sensory block** | | | | | | | |
| Lateral thigh at 3h | 17 (89.5) | 2 (10.5) | 0 (0) | 15 (83.3) | 3 (16.7) | 0 (0) | 0.66 |
| Lateral thigh at 6h | 18 (94.7) | 1 (5.3) | 0 (0) | 16 (88.9) | 2 (11.1) | 0 (0) | 0.60 |
| Lateral thigh at 24h | 19 (100) | 0 (0) | 0 (0) | 17 (94.4) | 1 (5.6) | 0 (0) | 0.49 |
| Anterior thigh at 3h | 15 (78.9) | 4 (21.1) | 0 (0) | 13 (72.2) | 4 (22.2) | 1 (5.6) | 0.84 |
| Anterior thigh at 6h | 16 (84.2) | 3 (15.8) | 0 (0) | 14 (77.8) | 4 (22.2) | 0 (0) | 0.47 |
| Anterior thigh at 24h | 18 (94.7) | 1 (5.3) | 0 (0) | 17 (94.4) | 1 (5.6) | 0 (0) | 0.74 |
| Medial thigh at 3h | 16 (84.2) | 3 (15.8) | 0 (0) | 14 (77.8) | 4 (22.2) | 0 (0) | 0.69 |
| Medial thigh at 6h | 17 (89.5) | 2 (10.5) | 0 (0) | 15 (83.3) | 3 (16.7) | 0 (0) | 0.47 |
| Medial thigh at 24h | 19 (0) | 0 (0) | 0 (0) | 16 (88.9) | 2 (11.1) | 0 (0) | 0.23 |
| **Post-operative motor block** | **No block** | **Paresis** | **Paral-ysis** | **No block** | **Paresis** | **Paral-ysis** | |
| Knee extension at 3h | 15 (78.9) | 2 (10.5) | 2 (10.5) | 14 (77.8) | 2 (11.1) | 2 (11.1) | >0.99 |
| Knee extension at 6h | 17 (89.5) | 1 (5,3) | 1 (5.3) | 14 (77.8) | 3 (16.7) | 1 (5,6) | 0.66 |
| Knee extension at 24h | 19 (100) | 0 (0) | 0 (0) | 17 (94.4) | 1 (5.6) | 0 (0) | 0.49 |

Data are expressed as number (proportion).

**Table 4. Post-operative outcomes and perioperative complications.**

| | Conventional vol-ume group (n = 19) | High volume group(n = 18) | Hazard ratio/median difference (95%CI) | *P* value |
|---|---|---|---|---|
| **Post-operative outcomes** | | | | |
| Time to first opioid, h(95%CI) | 9.5 (3.9–15.1)[a] | 5.3 (2.2–8.4)[a] | 0.59 (0.29,1.21)[c] | 0.15 |
| Post-operative opioids consumption, n(%) | | | | |
| Yes | 15 (78.9) | 17 (94.4) | – | 0.34 |
| No | 4 (21.1) | 1 (5.6) | – | – |
| Total consumption of opioids within 48h after operation (morphine equivalent: mg) | 10.13 (0.53,27.27)[b] | 10.13 (1.20,22.93)[b] | −0.22 (−1.23,0.89)[d] | 0.80 |
| Time to first walking (h) | 27.2 (24.3,32.3)[b] | 27.8 (24.3,48.4)[b] | −1.42 (−15.00,3.20)[d] | 0.61 |
| Length of hospital stay (d) | 5 (4,6)[b] | 6 (5,6)[b] | 0.00 (−1.00,0.00)[d] | 0.19 |
| **Perioperative complications** | | | | |
| Nausea | 1 (5.3) | 1 (5.6) | – | >0.99 |
| Vomiting | 1 (5.3) | 1 (5.6) | – | >0.99 |
| Dizziness | 0 (0) | 1 (5.6) | – | 0.49 |
| Pruritus | 1 (5.3) | 0 (0) | – | >0.99 |
| Respiratory depression | 0 (0) | 0 (0) | – | >0.99 |
| In- hospital falls | 0 (0) | 0 (0) | – | >0.99 |
| Wound infection | 0 (0) | 0 (0) | – | >0.99 |
| LAST | 0 (0) | 0 (0) | – | >0.99 |
| Vascular puncture | 0 (0) | 0 (0) | – | >0.99 |

Values are presented as median (95% CI), median (P25, P75), number (proportion), hazard ratio/median difference (95% CI), [a] refers to median (95% CI), [b] refers to median (P25, P75), [c] refers to hazard ratio (95% CI), [d] refers to median difference (95% CI). LAST, local anaesthetic systemic toxicity.

Ertaş G et al. [20] compared 30 ml and 20 ml PENG block in hemiarthroplasty for hip fracture, and found no difference in analgesic effect in positioning for spinal anesthesia and postoperative analgesia, which was consistent with our conclusion. We used a higher volume; however, within the constraints of this superiority trial, no statistically significant enhancement of anesthetic or analgesic effects was detected. Notably, quadriceps weakness was significantly greater in the PENG-30 group at 6 hours postoperatively in their study but it had resolved by the ninth hour. This difference may be related to surgical technique, nerve block proficiency, and other factors, with higher anesthetic volumes potentially accentuating these effects.

Leurcharusmee P [21] found that for PENG block, the MEV90 of methylene blue required to spare the femoral nerve and stain iliac bone between the AIIS and the IPE in a cadaveric model was 13.2 mL. Therefore, volumes exceeding 20 ml may result in a low proportion of femoral nerves spared, which is significantly different from our study. Contrary to findings from most *in vivo* studies, Girón-Arango *et al.* [8,9] found that 20 ml PENG block did not cause motor block, other scholars [22,23] found that the incidence of quadriceps weakness was about 10% at 6 h post-THA. One possible explanation is that the diffusion of local anesthetics in cadavers differs significantly from that in living bodies. Moreover, as observed by Leurcharusmee P *et al.*, the anatomical structure of cadavers undergoes considerable changes compared to living bodies. During performance of PENG block, the needle tip needs to be placed close to the femoral nerve. This close positioning makes it more likely that methylene blue will stain the femoral nerve along the needle's path, which in turn increases the chance of affecting the nerve.

Nevertheless, numerically longer times to first opioid use and a higher proportion of opioid-free patients were observed in the conventional-volume group, although these differences were not statistically significant. The possible reason for the findings is that, although there was no statistically significant difference between the groups in preoperative pain scores or in the proportion of patients not requiring opioids, the average preoperative dynamic pain score and the proportion of preoperative opioid use were higher in the high-volume group. Consequently, the high-volume PENG block did not effectively reverse the preoperative pain distribution in THA patients following the operation. There were no differences in other secondary outcomes such as sensory block and motor block; however, these analyses were exploratory, unadjusted for multiple comparisons, and may have been underpowered to detect modest or time-dependent effects.

Although the study achieved the prespecified sample size for the primary endpoint, the relatively small cohort limits the precision of effect estimates. Accordingly, the possibility of a type II error cannot be fully excluded, and the absence of statistically significant differences should be interpreted with caution. It should be noted that this study was designed as a superiority trial and was not powered or structured to formally assess equivalence or non-inferiority between treatment volumes.

Nonetheless, this study had several limitations. First, the dose of isobaric bupivacaine was 10 mg. Previous studies demonstrated that the median for acupuncture analgesia with 10 mg isobaric bupivacaine was 2 h (35–250 min) and the motor block median was 3 h (50–250 min) [24]. These data showed that spinal anesthesia would seriously affect the evaluation of sensorimotor block at 3 h post-operatively. However, spinal anesthesia is routinely preferred for THA in our institution. Therefore, future studies should consider using general anesthesia to allow for a more detailed and precise assessment of early post-operative sensory and motor block. Secondly, our assessment of lower limb muscle strength was affected by post-operative pain.For a more accurate assessment of the PENG block's impact on muscle strength, future research should consider involving healthy volunteers. Thirdly, in this study, assessors were blinded, while block performers were not, which may introduce potential performance bias.

## Conclusion

In this randomized superiority trial, high-volume PENG block did not demonstrate superior postoperative analgesia compared with conventional-volume PENG block in patients undergoing primary total hip arthroplasty. While no statistically significant differences were detected in pain scores, motor function, or opioid-related outcomes, the study was not

designed to establish equivalence between treatment volumes. Larger, adequately powered trials—ideally incorporating formal non-inferiority designs—are required to confirm these findings and to more precisely evaluate functional outcomes and uncommon adverse events.

## Supporting information

**S1 Checklist. CONSORT checklist.**
(DOCX)

**S2 File. Study protocol.**
(DOCX)

**S3 File. Raw data.**
(XLSX)

## Acknowledgments

The authors thank the patients for their participation in this study. The authors thank Dr. Huilin Zhang for his assistance with patient recruitment.

## Author contributions

**Conceptualization:** Zhao-hui Chen.

**Data curation:** Shi-ming Qin.

**Formal analysis:** Qian-song Wang, Zhao-hui Chen.

**Investigation:** Chong-mei Gao.

**Methodology:** Yang Zhao, Zhao-hui Chen.

**Project administration:** Xia Yuan.

**Supervision:** Zhao-hui Chen.

**Writing – original draft:** Qian-song Wang.

**Writing – review & editing:** Zhao-hui Chen.

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
