## [Decision Letter · Decision Letter 0]

7 Aug 2025

Dear Dr. Chen

Thank you for submitting your manuscript to PLOS ONE. After careful consideration, we feel that it has merit but does not fully meet PLOS ONE’s publication criteria as it currently stands. Therefore, we invite you to submit a revised version of the manuscript that addresses the points raised during the review process.

Thank you authors for the submission.

Please respond  point to point input from reviewers.

We look forward to receiving your revised manuscript.

Kind regards,

Rizaldy Taslim Pinzon

Academic Editor

PLOS ONE

Journal Requirements:

https://journals.plos.org/plosone/s/file?id=ba62/PLOSOne_formatting_sample_title_authors_affiliations.pdf ..

Additional Editor Comments (if provided):

Thank you authors for the submission.

Please respond point to point input from reviewers.

Reviewers' comments:

Reviewer's Responses to Questions

**Comments to the Author**

1. Is the manuscript technically sound, and do the data support the conclusions?

Reviewer #1: Yes

Reviewer #2: Yes

2. Has the statistical analysis been performed appropriately and rigorously?

Reviewer #1: Yes

Reviewer #2: Yes

3. Have the authors made all data underlying the findings in their manuscript fully available?

Reviewer #1: Yes

Reviewer #2: Yes

4. Is the manuscript presented in an intelligible fashion and written in standard English?

Reviewer #1: Yes

Reviewer #2: Yes

Reviewer #1: This is a well designed randomised trial to test the optimum volume PENG block for hip arthroplasty. I don't have any major comment on methodology and statistical analysis except the following minor comments.

1. Line 114: second was needs to be deleted

2. Line 224: Table 2; P value P should be italic.

3. P-values up to 2 decimal points are sufficient.

4. Multiple testing adjustment will make the secondary end points more non-significant and hence no need.

5. Table 4. time to first opioid hazards ratio seems to be 9.5 vs 5.3 but not significantly different. Please check this result. Although it is not significant; is the difference in right direction which needs to be commented.

Reviewer #2: Dear authors, thanks for the opportunity to revise your manuscript entitled "High-versus conventional-volume pericapsular nerve group (PENG) block for total hip arthroplasty: A randomized, controlled trial".

Following my comments:

- The results and discussion sections lack concise structuring. The narrative is repetitive in places, particularly where similar findings are discussed across multiple studies (e.g., Ahiskalioglu et al. and Girón-Arango et al.). Consider synthesizing this information to avoid redundancy and focus on the novel contribution of your study.

-The methods describing the PENG block procedure lack detail regarding ultrasound guidance, needle type, and exact positioning. These are essential for reproducibility. Additionally, specify the criteria for evaluating motor and sensory block (e.g., scale used, evaluator blinding, etc.).

-Sample size: State the software or method used for the power analysis (e.g., G*Power); Clearly specify the calculated sample size per group and total (e.g., 17 per group, 34 total);Justify the chosen minimum clinically important difference (MCID) clearly, noting it as a conservative estimate between moderate and severe pain.

-The conclusion that high-volume PENG block is not superior could be strengthened by discussing whether a type II error is possible due to sample size or variability in clinical performance. Some caution in the interpretation would increase credibility. Moreover, the limitations are acknowledged but could be expanded. For example, the influence of spinal anesthesia on early motor/sensory assessments is important and should be emphasized further. Additionally, the use of subjective pain scores without standardized rescue analgesia criteria should be discussed.

- Several grammatical issues and awkward phrases need editing.

-Ethical apporval? Registration of clinical trial?

-Please ad this important ciation on PENG block: DOI:10.3390/jcm13092674

**Do you want your identity to be public for this peer review?** For information about this choice, including consent withdrawal, please see our For information about this choice, including consent withdrawal, please see our Privacy Policy .

Reviewer #1: **Yes:** Dr Shah-Jalal SarkerDr Shah-Jalal Sarker

Reviewer #2: No

While revising your submission, please upload your figure files to the Preflight Analysis and Conversion Engine (PACE) digital diagnostic tool, https://pacev2.apexcovantage.com/ . PACE helps ensure that figures meet PLOS requirements. To use PACE, you must first register as a user. Registration is free. Then, login and navigate to the UPLOAD tab, where you will find detailed instructions on how to use the tool. If you encounter any issues or have any questions when using PACE, please email PLOS at . PACE helps ensure that figures meet PLOS requirements. To use PACE, you must first register as a user. Registration is free. Then, login and navigate to the UPLOAD tab, where you will find detailed instructions on how to use the tool. If you encounter any issues or have any questions when using PACE, please email PLOS at figures@plos.org . Please note that Supporting Information files do not need this step.. Please note that Supporting Information files do not need this step.

---

## [Author Response · Author response to Decision Letter 1]

11 Sep 2025

Manuscript No.: PONE-D-25-27762

Title: High-versus conventional-volume pericapsular nerve group (PENG) block for total hip arthroplasty: A randomized, controlled trial

Dear Prof Rizaldy Taslim Pinzon,

Thank you very much for your letter and helpful advice. We have revised the manuscript and would like to re-submit it for your consideration. We would like to express our sincere thanks to you and the reviewers for your constructive and positive comments which help us promote our manuscript a lot.

We have addressed all the comments raised by the editor and reviewers, and the amendments are highlighted in red in the revised manuscript. Besides, we checked the format according to your journal submission guidelines. Also, we have listed a point-by-point response to the editor and reviewers' comments as follows.

Your consideration of our revised manuscript for publication in your journal will be highly appreciated!

Best regards,

Zhao-Hui Chen,

Department of Anesthesiology,

Third Affiliated Hospital of Chongqing Medical University ,

No.1 Shuanghu Branch Road,

Chongqing, 401120, China;

Email: 650688@cqmu.edu.cn

Response to the reviewer 1

Manuscript No.: PONE-D-25-27762

Title: High-versus conventional-volume pericapsular nerve group (PENG) block for total hip arthroplasty: A randomized, controlled trial

Comment 1: Line 114: second was needs to be deleted.

Response: Thank you for your helpful and valuable comment. We have deleted unnecessary and lengthy expressions in brackets（line124-125）. Revision details were shown below

Clinical Trial Registry (ChiCTR, https://www.chictr.org.cn/, ID: ;ChiCTR2300077281;, Date of registration:November 3, 2023) before patient recruitment.

Comment 2: Line 224: Table 2; P value P should be italic.

Response: Thank you very much for your kind reminding. We have changed the P value in all tables to italics. Please check them in Table2-4 of the revised manuscript.

Comment 3: P-values up to 2 decimal points are sufficient.

Response: Thank you for your helpful and valuable comment. we have revised all P values with only 2 decimal points throughout the manuscript.

Comment 4: Multiple testing adjustment will make the secondary end points more non-significant and hence no need.

Response: Thank you for your helpful and valuable comment. We deleted the dynamic and static VAS pain scores at 48 hours after surgery. Please check them in Table2 of the revised manuscript. Related descriptions have also been removed.(line59,line115,line174-175)

Comment 5: Table 4. time to first opioid hazards ratio seems to be 9.5 vs 5.3 but not significantly different. Please check this result. Although it is not significant; is the difference in right direction which needs to be commented.

Response: Thank you for your helpful and valuable comment. We checked the original data and statistical process, and have supplemented the discussion of these results, and the details are shown below(Discussion section, line 312-322).

There was no significant difference between the two groups in time to first opioid use or in the proportion of patients who did not require opioids within 48 hours after operation. However, the conventional volume group showed a trend toward a later time to the first opioid use and a higher proportion of patients without opioid use. The possible reason was that although there was no statistically significant difference between the groups in preoperative pain scores or the proportion of patients not requiring opioids. the average preoperative dynamic pain score and the proportion of preoperative opioid use were higher in the high-volume group. The high-volume PENG block failed to reverse the preoperative pain distribution in THA patients after operation. Furthermore, as our sample size reached the pre-specified target, the likelihood of a type II error in our conclusion is very low, which further supports that the postoperative analgesic effect of high-volume PENG block is not superior to that of the conventional volume group.

Response to the reviewer 2

Manuscript No.: PONE-D-25-27762

Title: High-versus conventional-volume pericapsular nerve group (PENG) block for total hip arthroplasty: A randomized, controlled trial

Comment 1: The results and discussion sections lack concise structuring. The narrative is repetitive in places, particularly where similar findings are discussed across multiple studies (e.g., Ahiskalioglu et al. and Girón-Arango et al.). Consider synthesizing this information to avoid redundancy and focus on the novel contribution of your study.

Response:Thank you for your helpful and valuable comment. We have greatly revised the results （line227-236,line241-252）and discussions section (line257-265,line266-288)., trying to be concise, particularly where similar findings are discussed across multiple studies (e.g., Ahiskalioglu et al. and Girón-Arango et al.)。 Details were shown below:

The dynamic pain scores of the high- and conventional-volume groups at 6 h after surgery were 5.0(4.0,6.0) and 4.0(4.0,5.0), respectively, indicating that their dynamic pain scores showed no statistically significant differences (Table 2). No significant differences in the dynamic pain scores were found at 3 h and 24 h between the two groups. Meanwhile, no statistically significant differences in the static pain scores were observed between the two groups during all study time points (Table 2). There was no significant difference in the dermatomal sensory block of the lateral, anterior and medial thigh on the surgical side between the two groups. Similarly, no significant differences were observed in the motor block on the surgical side between the two groups (Table 3). There was no significant differences were observed between the two groups in dynamic pain scores at 3 and 24 hours, static pain scores at any time point, dermatomal sensory block, or motor block on the surgical side. (Table 2,3).

Four patients (21.1%) in the conventional-volume group and one patient (5.6%) in the high-volume group did not require any opioids for 48 h. No statistically significant differences in the proportion of opioid requirements, time of primary opioid consumption, and total opioid consumption within 48 h were found between the two groups (Table 4). No statistically significant differences in the time of first walking and length of hospital stay were observed between the two groups. No statistically significant differences in the incidence of post-operative opioid-related complications (e.g., nausea, vomiting, pruritus, and dizziness) were found between the two groups (Table 4). There was no significant differences in the proportion of opioid requirements, time of primary opioid consumption, total opioid consumption within 48 h, time of first walking , length of hospital stay and incidence of post-operative opioid-related complications were found between the two groups (Table 4). Post-operative respiratory depression, in-hospital falls, wound infections, intravascular injections, and local anesthetic poisoning were observed in neither group during nerve block (Table 4).

In this randomized trial, we compared high-volume versus conventional-volume PENG blocks in patients undergoing primary THA. The results revealed no statistically significant differences in the dynamic pain scores between the two groups at 6 h post-operatively. At 6 h post-operatively, the two groups showed no statistically significant difference in dynamic pain scores. Similarly, no statistically significant differences in the dynamic pain score at other time-points and the static pain scores at all time-points were found. Furthermore, dynamic pain scores at other time points and static pain scores throughout the entire period were statistically similar between the groups. In addition, no statistically significant differences there were no significant differences in the dermatomal sensory block and motor block were found at all time-points post-operatively.

Ahiskalioglu et al. found that the high-volume PENG block showed the characteristics of a lumbar plexus block and the innervation areas of the obturator nerve, the femoral nerve, and the lateral femoral cutaneous nerve were sufficiently anesthetized. [13, 18]. Given that long-acting local anesthetics used in nerve blocks provide post-operative analgesia lasting over 10 h, the pain scores within the first 0–6 h post-operatively are expected to be significantly lower with high-volume PENG blocks compared to conventional-volume PENG blocks, which was inconsistent with the results of the present study. However, it should be noted that quadriceps weakness occurs more frequently with high-volume PENG blocks. Nevertheless, the results of the present study are inconsistent with the trend reported by Ahiskalioglu et al. In the study by Ahiskalioglu et al, during the application of PENG block, the needle tip was located on the medial side of the iliopsoas tendon, and the local anesthetic spread along the psoas major and pubic muscles, thus blocking the obturator nerve, the lateral femoral cutaneous nerve and the femoral nerve. Therefore, the lumbar plexus block-like characteristics of the high-volume PENG block observed by Ahiskalioglu et al. may be related to suboptimal PENG block due to the aforementioned reasons. Elsewhere, Girón-Arango et al[15, 19], inferred that another reason for the unexpected quadriceps myasthenia or the appearance similar to the lumbar plexus block in the PENG block was that the local anesthetic injection was too superficial. In this study, the PENG block was performed under intravenous analgesia and sedation, and subcutaneous local anesthetic injections were contraindicated. The tip of the needle was required to reach the bone surface of the iliopubic eminence and stick to the lateral side of the iliopsoas tendon before injecting local anesthetic. to avoid accidental superficial injection and unexpected diffusion of local anesthetic. We found that the standard PENG block did not block the peripheral branches of lumbar plexus even with high volume Our results confirm that high-volume PENG block does not block the obturator nerve effectively and will not increase the analgesic effect.

Comment 2: The methods describing the PENG block procedure lack detail regarding ultrasound guidance, needle type, and exact positioning. These are essential for reproducibility. Additionally, specify the criteria for evaluating motor and sensory block (e.g., scale used, evaluator blinding, etc.)

Response:Thank you for your helpful and valuable comment.

1.In the part of materials and methods, we described the whole procedure of ultrasound-guided PENG block, including the types and positioning methods of needles（Line151-162）and added a figure（figure 1） of ultrasound-guided PENG block. Please check it.Details were shown below:

Performance of PENG blocks

The ultrasound-guided PENG block was performed preoperatively, with routine monitoring of blood pressure, oxygen saturation, and electrocardiogram; the patient was placed in the supine position. The ultrasound convex array probe (2-5 Hz) (HITACHI ALOKA ARIETTA, Fujifilm Medical, Tokyo, Japan) was positioned on transverse orientation, medial and caudal to the anterosuperior iliac spine to identify the anteroinferior iliac spine, the iliopubic eminence and the psoas tendon. Using an in-plane technique and a lateral to medial orientation, the block needle (B. Braun Melsungen AG, Melsungen ，Germany) was advanced until its tip was located on the periosteum on the dorsal side of the iliopsoas tendon. After negative aspiration, the high-volume group was injected with 40 mL of 0.375% ropivacaine, and the conventional-volume group received 20 mL of 0.375% ropivacaine, resulting in a local anesthetic between the periosteum of the iliopsoas tendon and iliopubic eminence(Figure 1). Dexamethasone (5mg) was mixed with the local anesthetic and administered simultaneously in each group.

2.In the materials and methods section, we introduced the methods of evaluating motor and sensory block in detail(line181-189). Details were shown below:

Post-operative sensory block was evaluated in the anterior, lateral and medial aspects of the mid-thigh innervated by the femoral nerve, lateral femoral cutaneous nerve, obturator and femoral nerve respectively at 3 h, 6 h, and 24 h post-surgery using the Aliste J's method. For each region, the blockade was evaluated on a 3-point scale: 2 = no block, 1 = analgesia (the patient can feel touch, not cold), 0 = anesthesia (patient can't feel touch). Post-operative motor block was investigated through knee extension, which was examined while in supine position with the patient’s hip and knee flexed at 45° and 90°, respectively. The patient was asked to extend the knee against gravity and resistance, the extension was graded based on a 3- point scale: 2=no block (extension against gravity and against resistance); 1= paresis (extension against gravity but not against resistance); 0=paralysis (no extension possible).

3. In the materials and methods , we introduced the blind method of postoperative sensory and motor block evaluation in the materials and methods(line141-150). Details were shown below:

A nurse anesthetist who was not involved in the study randomly divided patients into two groups with approximately equal sample sizes using a random number table in a 1:1 ratio: the high-volume group (even numbers) and the conventional-volume group (odd numbers). The grouping data were sealed in sequentially numbered opaque envelopes, which were opened on the day of surgery by researchers who performed the nerve block. All nerve blocks were performed by a trained anesthesiologist who was proficient in PENG blocks. Another anesthesiologist from the acute pain service team who was not involved in the nerve block procedure or intraoperative management and was blinded to the grouping of the patients followed up with the patients after surgery to assess post-operative pain scores and sensory and motor blocks and collect data. Unblinding was performed once the data collection process was completed for all patients.

Comment 3: Sample size: State the software or method used for the power analysis (e.g., G*Power); Clearly specify the calculated sample size per group and total (e.g., 17 per group, 34 total);Justify the chosen minimum clinically important difference (MCID) clearly, noting it as a conservative estimate between moderate and severe pain.

Response: Thank you for your kind reminder and comment. In the part of sample size estimation, we added the name of the software used for sample size estimation. At the same time, we described why we chose 1.5 as the minimum dynamic pain score reduction value with clinical significance, and described how to calculate the sample size of each group and the total sample size(line194-206). Details were shown below:

Sample size calculation was performed using PASS version 11(Power Analysis and Sample Size Software).The minimum decrease clinically important difference in pain scores that is considered meaningful to patients is associated with their was determined based on patients’ baseline pain levels. For individuals experiencing moderate pain, a change of 1.3 on the VAS is the threshold for minimum clinically significant pain relief. In contrast, for those with severe baseline pain, a reduction of 1.8 on the VAS is deemed clinically significant[16]. A recent clinical trial reported that the dynamic pain score at 6 h after surgery in patients with a PENG block was 6[11], which was considered between moderate and severe pain. Therefore, we chose 1.5 as the clinically meaningful minimum dynamic pain score reduction for patients receiving the PENG block. Based on previous data on the use of the PENG block in THA[9, 17], the standard deviation (SD) was estimated to be 1.5. To test the difference in pain score of 1.5 (out of 10) and SD of 1.5, a two-tailed, independent samples t-test with an α-error of 0.05 and a β-error of 0.2 was conducted; a minimum of 17 patients were required in each group. Considering the imbalance of grouping and drop-outs, 40 patients were recruited.

Comment 4: The conclusion that

---

## [Decision Letter · Decision Letter 1]

3 Dec 2025

Dear Dr. Chen,

Thank you for submitting your manuscript to PLOS ONE. After careful consideration, we feel that it has merit but does not fully meet PLOS ONE’s publication criteria as it currently stands. Therefore, we invite you to submit a revised version of the manuscript that addresses the points raised during the review process.

We look forward to receiving your revised manuscript.

Kind regards,

Nan Jiang

Academic Editor

PLOS ONE

Journal Requirements:

Reviewers' comments:

Reviewer's Responses to Questions

**Comments to the Author**

Reviewer #1: All comments have been addressed

2. Is the manuscript technically sound, and do the data support the conclusions?

Reviewer #1: Yes

3. Has the statistical analysis been performed appropriately and rigorously?

Reviewer #1: (No Response)

4. Have the authors made all data underlying the findings in their manuscript fully available?

Reviewer #1: Yes

5. Is the manuscript presented in an intelligible fashion and written in standard English?

Reviewer #1: Yes

Reviewer #1: Manuscript title: High- versus conventional-volume pericapsular nerve group (PENG) block for total hip arthroplasty: A randomized, controlled trial

Manuscript number: PONE-D-25-27762R1

General Assessment

The authors have addressed the prior reviewer comments carefully, and the revised version presents a more transparent and methodologically complete trial report. The inclusion of detailed methods for the block procedure, motor/sensory assessment criteria, randomization, blinding, sample size justification, and trial registration significantly strengthens the manuscript. The statistical reporting is improved, though some limitations remain. Overall, this is a well-conducted small RCT, but some issues around interpretation, reporting, and methodological clarity should be noted.

Major Points

1. Primary Outcome & Power Analysis

o The primary outcome (dynamic VAS at 6h) is clearly defined and analysed.

o The power calculation is now adequately justified, with an MCID of 1.5 points on the VAS and SD = 1.5. The use of PASS software is appropriate.

o However, the total analysed sample (n=37) is only marginally above the minimum of 34 required. Although dropouts were low, the risk of type II error cannot be entirely excluded. The authors argue that it is unlikely, but more cautious interpretation would be preferable.

2. Statistical Methods

o Analyses are appropriate: t-test/Mann-Whitney U for continuous variables, χ²/Fisher’s exact test for categorical data, log-rank test for time-to-event outcomes.

o P-values are now standardized to two decimal places. Confidence intervals are presented for time-to-event data but could be more consistently reported for other outcomes to improve interpretability.

o Adjustment for multiple testing was deliberately not performed, which is acceptable since secondary endpoints are exploratory, but this should be explicitly acknowledged in the Discussion to avoid misinterpretation.

3. Randomization & Blinding

o Randomization method (opaque envelopes with 1:1 allocation) is appropriate.

o Blinding is well described: block performers were not blinded, but assessors were. The potential for performance bias should be acknowledged more clearly.

4. Outcome Assessments

o Sensory and motor block evaluation scales are now clearly described and reproducible.

o The influence of spinal anaesthesia on early block assessment is discussed, but the limitation remains important. At 3h post-op, residual spinal effect could confound sensory/motor findings. This undermines the reliability of early block outcome measures.

5. Results & Interpretation

o No statistically significant differences were found between groups. The Discussion appropriately cites possible explanations (baseline imbalance in opioid use, volume-related spread inconsistencies).

o However, the interpretation could still overstate certainty. While the trial is adequately powered for the primary endpoint, the sample size is small for detecting uncommon complications or subtle functional differences.

6. Transparency & Reporting

o CONSORT flow diagram and trial registration are now included.

o Ethical approval and informed consent are documented.

o Data availability statement is satisfactory.

o The manuscript is clearer after language editing, though minor stylistic inconsistencies remain.

Minor Points

• Tables should consistently report both effect size estimates (mean/median difference) and 95% CI, not only P-values.

• In Table 4, the trend toward later time-to-first opioid in the conventional group is acknowledged in the Discussion, but presenting hazard ratios with CI would better quantify this.

• The claim that a “type II error is very unlikely” should be softened to reflect uncertainty.

• The conclusion would be more balanced if it emphasized that conventional volume is sufficient, rather than definitively “optimal.”

Recommendation

The study is methodologically sound, and the revisions have substantially improved clarity and reproducibility. Remaining issues are relatively minor and relate mainly to cautious interpretation and strengthening statistical reporting (confidence intervals, effect sizes).

I recommend acceptance after minor revision, focusing on:

1. Consistent reporting of confidence intervals alongside P-values.

2. Softer language regarding type II error and the “optimal” dose conclusion.

3. Explicit note that secondary outcomes were exploratory and unadjusted for multiplicity.

**Do you want your identity to be public for this peer review?** For information about this choice, including consent withdrawal, please see our For information about this choice, including consent withdrawal, please see our Privacy Policy .

Reviewer #1: **Yes:** Dr Shah-Jalal SarkerDr Shah-Jalal Sarker

---

## [Author Response · Author response to Decision Letter 2]

17 Dec 2025

Manuscript No.: PONE-D-25-27762R1

Title: High-versus conventional-volume pericapsular nerve group (PENG) block for total hip arthroplasty: A randomized, controlled trial

Dear Dr Nan Jiang,

Thank you very much for your letter and helpful advice. We have revised the manuscript and would like to re-submit it for your consideration. We would like to express our sincere thanks to you and the reviewers for your constructive and positive comments which help us promote our manuscript a lot.

We have addressed all the comments raised by the editor and reviewers, and the amendments are highlighted in red in the revised manuscript. Besides, We checked all the references and found that none of the cited references had been withdrawn.. Also, we have listed a point-by-point response to the editor and reviewers' comments as follows.

Your consideration of our revised manuscript for publication in your journal will be highly appreciated!

Best regards,

Zhao-Hui Chen,

Department of Anesthesiology,

Third Affiliated Hospital of Chongqing Medical University ,

No.1 Shuanghu Branch Road,

Chongqing, 401120, China;

Email: 650688@cqmu.edu.cn

Response to the reviewer

Manuscript No.: PONE-D-25-27762R1

Title: High-versus conventional-volume pericapsular nerve group (PENG) block for total hip arthroplasty: A randomized, controlled trial

Response to major points

Comment 1:

Primary Outcome & Power Analysis

o The primary outcome (dynamic VAS at 6h) is clearly defined and analysed.

o The power calculation is now adequately justified, with an MCID of 1.5 points on the VAS and SD = 1.5. The use of PASS software is appropriate.

o However, the total analysed sample (n=37) is only marginally above the minimum of 34 required. Although dropouts were low, the risk of type II error cannot be entirely excluded. The authors argue that it is unlikely, but more cautious interpretation would be preferable.

Response: Thank you for your helpful and valuable comment. We have revised the description of type II errors,（line304-307）. Revision details were shown below:

Furthermore, as our sample size has reached the pre-specified target, the likelihood of a type II error in our conclusion is very low,which reduces the probability of type II errors, which further supports that the postoperative analgesic effect of high-volume PENG block is not superior to that of the conventional volume group.

Comment 2:

Statistical Methods

o Analyses are appropriate: t-test/Mann-Whitney U for continuous variables, χ²/Fisher’s exact test for categorical data, log-rank test for time-to-event outcomes.

o P-values are now standardized to two decimal places. Confidence intervals are presented for time-to-event data but could be more consistently reported for other outcomes to improve interpretability.

o Adjustment for multiple testing was deliberately not performed, which is acceptable since secondary endpoints are exploratory, but this should be explicitly acknowledged in the Discussion to avoid misinterpretation.

Response: Thank you very much for your helpful and valuable comment. We increased the median difference and risk ratio and the 95% confidence interval. Please check them in Table2 and 4 of the revised manuscript. We have added the discussion of repeated measurement data(line319-320), revision details were shown below:

Additionally, some secondary endpoints involved repeated measurement data; however, adjustment for multiple testing was not performed since these endpoints were exploratory.

Comment 3:

Randomization & Blinding

o Randomization method (opaque envelopes with 1:1 allocation) is appropriate.

o Blinding is well described: block performers were not blinded, but assessors were. The potential for performance bias should be acknowledged more clearly.

Response: Thank you for your helpful and valuable comment. We have added a discussion about the limitations of blind method(line318-319), revision details were shown below:

Thirdly, in this study, assessors were blinded, while block performers were not, which may introduce potential performance bias.

Comment 4:

Outcome Assessments

o Sensory and motor block evaluation scales are now clearly described and reproducible.

o The influence of spinal anaesthesia on early block assessment is discussed, but the limitation remains important. At 3h post-op, residual spinal effect could confound sensory/motor findings. This undermines the reliability of early block outcome measures.

Response: Thank you for your helpful and valuable comment. Please check it in line 308-315 for the related limitation description

Comment 5:

Results & Interpretation

o No statistically significant differences were found between groups. The Discussion appropriately cites possible explanations (baseline imbalance in opioid use, volume-related spread inconsistencies).

o However, the interpretation could still overstate certainty. While the trial is adequately powered for the primary endpoint, the sample size is small for detecting uncommon complications or subtle functional differences.

Response: Thank you for your helpful and valuable comment. In the discussion of limitations,we added a description of the influence of small sample size on secondary results,and the details are shown below(Discussion section, line 321-322).

Fourthly, although the sample size was adequate for the primary endpoint, it was relatively small and may not have been sufficient to detect differences in the secondary endpoints.

Comment 6:

Transparency & Reporting

o CONSORT flow diagram and trial registration are now included.

o Ethical approval and informed consent are documented.

o Data availability statement is satisfactory.

o The manuscript is clearer after language editing, though minor stylistic inconsistencies remain.

Response: Thank you for your helpful and valuable comment. We invited a retouching organization to polish the full text carefully again. Please check the details of these changes carefully (line88-93,line104,106-111,line1139-146,line159-161,line232,line241,line244,line253-258,line260-270,line295-303)

Response to minor Points

Comment 1: Tables should consistently report both effect size estimates (mean/median difference) and 95% CI, not only P-values

Response:Thank you for your helpful and valuable comment. We added the median difference and 95% confidence interval in Table 2 and 4. Please check them

Comment 2: In Table 4, the trend toward later time-to-first opioid in the conventional group is acknowledged in the Discussion, but presenting hazard ratios with CI would better quantify this.

Response:Thank you for your helpful and valuable comment. We have added the risk ratio and 95% confidence interval in Table 4. Please check it.

Comment 3: The claim that a “type II error is very unlikely” should be softened to reflect uncertainty.

Response: Thank you for your kind reminder and comment. We have revised the relevant description(line304-7), and the details are as follows:

Furthermore, as our sample size has reached the pre-specified target, the likelihood of a type II error in our conclusion is very low,which reduces the probability of type II errors, which further supports that the postoperative analgesic effect of high-volume PENG block is not superior to that of the conventional volume group.

Comment 4: The conclusion would be more balanced if it emphasized that conventional volume is sufficient, rather than definitively “optimal.”

Response: Thank you for your helpful and valuable comment. We have revised the relevant description(line327-330), and the details are as follows:

Consequently, the conventional volume appears to be the optimal dose of PENG block for Therefore, the conventional volume is sufficient for PENG block in hip and lower limb surgery, and we do not recommend increasing the volume because wrong or nonstandard PENG block can elevate the risk of quadriceps femoris weakness, and a high volume will amplify this effect. Therefore, the conventional volume is sufficient for PENG block.

Response to key points in recommendation

Comment

1. Consistent reporting of confidence intervals alongside P-values.

2. Softer language regarding type II error and the “optimal” dose conclusion.

3. Explicit note that secondary outcomes were exploratory and unadjusted for multiplicity.

Response: Thank you for your helpful and valuable comment. We have responded to these points in the major points and minor points above. Please check them

---

## [Decision Letter · Decision Letter 2]

28 Dec 2025

Dear Dr. Chen,

Thank you for submitting your manuscript to PLOS ONE. After careful consideration, we feel that it has merit but does not fully meet PLOS ONE’s publication criteria as it currently stands. Therefore, we invite you to submit a revised version of the manuscript that addresses the points raised during the review process.

We look forward to receiving your revised manuscript.

Kind regards,

Nan Jiang

Academic Editor

PLOS One

Journal Requirements:

Reviewers' comments:

Reviewer's Responses to Questions

**Comments to the Author**

Reviewer #1: (No Response)

2. Is the manuscript technically sound, and do the data support the conclusions?

Reviewer #1: Partly

3. Has the statistical analysis been performed appropriately and rigorously?

Reviewer #1: Yes

4. Have the authors made all data underlying the findings in their manuscript fully available?

Reviewer #1: Yes

5. Is the manuscript presented in an intelligible fashion and written in standard English?

Reviewer #1: Yes

Reviewer #1: The authors have made a sincere effort to address prior reviewer feedback, and the manuscript has improved in clarity and reporting. However, interpretation of negative findings remains overstated, particularly with respect to type II error, equivalence inference, and practice recommendations. Additional softening is required to align the manuscript with accepted standards for superiority trials reporting null results.

The concerns outlined below are primarily interpretive, not methodological, and can be resolved with targeted language revision.

1. Type II Error: Language Remains Overconfident

Reviewer Concern (Persisting)

Despite prior feedback, the manuscript continues to assert that the likelihood of a type II error is “very low,” and uses this assertion to support conclusions of non-superiority. This is not statistically justified given:

• Small total sample size (n = 37)

• Power calculation limited to a single primary endpoint

• Absence of equivalence or non-inferiority design

Merely replacing “unlikely” with “very low” does not meaningfully soften the claim.

Why Further Softening Is Required

• A superiority trial that fails to detect a difference cannot rule out clinically meaningful effects.

• Power calculations do not eliminate type II error; they only constrain it under ideal assumptions.

• Repeated emphasis on “low type II error” risks misleading readers into inferring equivalence.

Required Change (Discussion)

Current text (representative):

“…our sample size has reached the pre-specified target, which reduces the probability of type II errors, which further supports that the postoperative analgesic effect of high-volume PENG block is not superior…”

Recommended replacement text:

“Although the study achieved the prespecified sample size for the primary endpoint, the relatively small cohort limits the precision of effect estimates. Accordingly, the possibility of a type II error cannot be fully excluded, and the absence of statistically significant differences should be interpreted with caution.”

2. Superiority vs. Equivalence: Inference Drift

Reviewer Concern

The trial is explicitly designed as a superiority trial, yet the Discussion and Conclusion increasingly imply equivalence or sufficiency, without appropriate methodological justification.

Why Further Softening Is Required

• Equivalence or non-inferiority cannot be inferred from a non-significant superiority test.

• Statements suggesting that conventional volume is “sufficient” or “optimal” exceed the evidentiary scope of the trial.

Required Change (Discussion)

Add the following clarifying sentence near the limitations paragraph:

“It should be noted that this study was designed as a superiority trial and was not powered or structured to formally assess equivalence or non-inferiority between treatment volumes.”

3. Secondary Outcomes and Multiplicity: Interpretation Still Too Strong

Reviewer Concern

While the authors now acknowledge that secondary outcomes were exploratory and unadjusted for multiplicity, interpretive language remains confirmatory.

Why Further Softening Is Required

• Multiple secondary endpoints increase false-negative and false-positive risk.

• Repeated-measures outcomes were analysed using independent comparisons, further limiting inference strength.

Required Change (Discussion)

Current text (representative):

“No significant differences were observed in dermatomal sensory block and motor block…”

Recommended replacement text:

“No statistically significant differences were observed in secondary outcomes; however, these analyses were exploratory, unadjusted for multiple comparisons, and may have been underpowered to detect modest or time-dependent effects.”

4. Conclusion: Practice Recommendations Are Premature

Reviewer Concern (Major)

The Conclusion currently makes clinical recommendations and suggests sufficiency of conventional volume, despite acknowledged limitations.

Why Further Softening Is Required

• PLOS ONE discourages definitive practice recommendations from small, single-centre trials.

• Conclusions should reflect what was tested, not what is implied.

Required Replacement Text: Conclusion (Full)

Current conclusion (summarized): “Therefore, the conventional volume is sufficient… we do not recommend increasing the volume…”

Recommended revised conclusion (replace entire section):

“In this randomized superiority trial, high-volume PENG block did not demonstrate superior postoperative analgesia compared with conventional-volume PENG block in patients undergoing primary total hip arthroplasty. While no statistically significant differences were detected in pain scores, motor function, or opioid-related outcomes, the study was not designed to establish equivalence between treatment volumes. Larger, adequately powered trials—ideally incorporating formal non-inferiority designs—are required to confirm these findings and to more precisely evaluate functional outcomes and uncommon adverse events.”

**Do you want your identity to be public for this peer review?** For information about this choice, including consent withdrawal, please see our For information about this choice, including consent withdrawal, please see our Privacy Policy .

Reviewer #1: **Yes:** Dr Shah-Jalal SarkerDr Shah-Jalal Sarker

---

## [Author Response · Author response to Decision Letter 3]

6 Jan 2026

Dear Dr Nan Jiang,

Thank you very much for your letter and helpful advice. We have revised the manuscript and would like to re-submit it for your consideration. We would like to express our sincere thanks to you and the reviewers for your constructive and positive comments which help us promote our manuscript a lot.

We have addressed all the comments raised by the editor and reviewers, and the amendments are highlighted in red in the revised manuscript. Besides, we have adjusted some paragraphs.Also, we have listed a point-by-point response to the editor and reviewers' comments as follows.

Your consideration of our revised manuscript for publication in your journal will be highly appreciated!

Best regards,

Zhao-Hui Chen,

Department of Anesthesiology,

Third Affiliated Hospital of Chongqing Medical University ,

No.1 Shuanghu Branch Road,

Chongqing, 401120, China;

Email: 650688@cqmu.edu.cn

Response to the reviewer

Manuscript No.: PONE-D-25-27762R2

Title: High-versus conventional-volume pericapsular nerve group (PENG) block for total hip arthroplasty: A randomized, controlled trial

Comment 1:

Type II Error: Language Remains Overconfident

Reviewer Concern (Persisting)

Despite prior feedback, the manuscript continues to assert that the likelihood of a type II error is “very low,” and uses this assertion to support conclusions of non-superiority. This is not statistically justified given:

• Small total sample size (n = 37)

• Power calculation limited to a single primary endpoint

• Absence of equivalence or non-inferiority design

Merely replacing “unlikely” with “very low” does not meaningfully soften the claim.

Why Further Softening Is Required

• A superiority trial that fails to detect a difference cannot rule out clinically meaningful effects.

• Power calculations do not eliminate type II error; they only constrain it under ideal assumptions.

• Repeated emphasis on “low type II error” risks misleading readers into inferring equivalence.

Required Change (Discussion)

Current text (representative):

“…our sample size has reached the pre-specified target, which reduces the probability of type II errors, which further supports that the postoperative analgesic effect of high-volume PENG block is not superior…”

Recommended replacement text:

“Although the study achieved the prespecified sample size for the primary endpoint, the relatively small cohort limits the precision of effect estimates. Accordingly, the possibility of a type II error cannot be fully excluded, and the absence of statistically significant differences should be interpreted with caution.”

Response: Thank you for your helpful and valuable comment. We have revised the description of type II errors and adopted the description of the reviewer（line292-294,line298-301）. Revision details were shown below:

Furthermore, our sample size has reached the pre-specified target, which reduces the probability of type II errors, which further supports that the postoperative analgesic effect of high-volume PENG block is not superior to that of the conventional volume group.(line292-294).

Although the study achieved the prespecified sample size for the primary endpoint, the relatively small cohort limits the precision of effect estimates. Accordingly, the possibility of a type II error cannot be fully excluded, and the absence of statistically significant differences should be interpreted with caution.(line298-301)

Comment 2:

Superiority vs. Equivalence: Inference Drift

Reviewer Concern

The trial is explicitly designed as a superiority trial, yet the Discussion and Conclusion increasingly imply equivalence or sufficiency, without appropriate methodological justification.

Why Further Softening Is Required

• Equivalence or non-inferiority cannot be inferred from a non-significant superiority test.

• Statements suggesting that conventional volume is “sufficient” or “optimal” exceed the evidentiary scope of the trial.

Required Change (Discussion)

Add the following clarifying sentence near the limitations paragraph:

“It should be noted that this study was designed as a superiority trial and was not powered or structured to formally assess equivalence or non-inferiority between treatment volumes.”

Response: Thank you very much for your helpful and valuable comment. We have deleted the description that conventional volume is sufficient(line323) , modified some sentences to soften the conclusion(line263-264,line267-268) and added the clarifying sentence near the limitations paragraph. (line301-302), revision details were shown below:

Therefore, the conventional volume is sufficient for PENG block in hip and lower limb surgery,

We have not found that the standard PENG block did not can block the peripheral branches of lumbar plexus even with high volume.

We used a higher volume, which may further confirmed strengthen that the high-volume PENG block did not enhance anesthetic or analgesic effects.

It should be noted that this study was designed as a superiority trial and was not powered or structured to formally assess equivalence or non-inferiority between treatment volumes.

Comment 3:

Secondary Outcomes and Multiplicity: Interpretation Still Too Strong

Reviewer Concern

While the authors now acknowledge that secondary outcomes were exploratory and unadjusted for multiplicity, interpretive language remains confirmatory.

Why Further Softening Is Required

• Multiple secondary endpoints increase false-negative and false-positive risk.

• Repeated-measures outcomes were analysed using independent comparisons, further limiting inference strength.

Required Change (Discussion)

Current text (representative):

“No significant differences were observed in dermatomal sensory block and motor block…”

Recommended replacement text:

“No statistically significant differences were observed in secondary outcomes; however, these analyses were exploratory, unadjusted for multiple comparisons, and may have been underpowered to detect modest or time-dependent effects.”

Response: Thank you for your helpful and valuable comment. We have revised the description of secondary outcomes and multiplicity. (line249-250,line315-318,line294-297), revision details were shown below:

Additionally, no significant differences were observed in dermatomal sensory block and motor block at all postoperative time points. .(lne249-250)

Additionally, some secondary endpoints involved repeated measurement data; however, adjustment for multiple testing was not performed since these endpoints were exploratory. Fourthly, although the sample size was adequate for the primary endpoint, it was relatively small and may not have been sufficient to detect differences in the secondary endpoints. (line315-318)

There were no differences in secondary outcomes such as sensory block and motor block; however, these analyses were exploratory, unadjusted for multiple comparisons, and may have been underpowered to detect modest or time-dependent effects.(line294-297)

Comment 4:

Conclusion: Practice Recommendations Are Premature

Reviewer Concern (Major)

The Conclusion currently makes clinical recommendations and suggests sufficiency of conventional volume, despite acknowledged limitations.

Why Further Softening Is Required

• PLOS ONE discourages definitive practice recommendations from small, single-centre trials.

• Conclusions should reflect what was tested, not what is implied.

Required Replacement Text: Conclusion (Full)

Current conclusion (summarized): “Therefore, the conventional volume is sufficient… we do not recommend increasing the volume…”

Recommended revised conclusion (replace entire section):

“In this randomized superiority trial, high-volume PENG block did not demonstrate superior postoperative analgesia compared with conventional-volume PENG block in patients undergoing primary total hip arthroplasty. While no statistically significant differences were detected in pain scores, motor function, or opioid-related outcomes, the study was not designed to establish equivalence between treatment volumes. Larger, adequately powered trials—ideally incorporating formal non-inferiority designs—are required to confirm these findings and to more precisely evaluate functional outcomes and uncommon adverse events.”

Response: Thank you for your helpful and valuable comment. We revised the conclusion and adopted the description of the reviewer(Line320-325,line326-332), revision details were shown below:

This study found that the high-volume PENG block achieved similar post-operative analgesic performance to the conventional-volume PENG block for primary THA. In addition, the incidence of quadriceps myasthenia of lower limbs was comparable to that of conventional PENG block. Therefore, the conventional volume is sufficient for PENG block in hip and lower limb surgery, and we do not recommend increasing the volume because wrong or nonstandard PENG block can elevate the risk of quadriceps femoris weakness, and a high volume will amplify this effect.

In this randomized superiority trial, high-volume PENG block did not demonstrate superior postoperative analgesia compared with conventional-volume PENG block in patients undergoing primary total hip arthroplasty. While no statistically significant differences were detected in pain scores, motor function, or opioid-related outcomes, the study was not designed to establish equivalence between treatment volumes. Larger, adequately powered trials—ideally incorporating formal non-inferiority designs—are required to confirm these findings and to more precisely evaluate functional outcomes and uncommon adverse events.

---

## [Decision Letter · Decision Letter 3]

20 Jan 2026

Dear Dr. Chen,

Thank you for submitting your manuscript to PLOS ONE. After careful consideration, we feel that it has merit but does not fully meet PLOS ONE’s publication criteria as it currently stands. Therefore, we invite you to submit a revised version of the manuscript that addresses the points raised during the review process.

We look forward to receiving your revised manuscript.

Kind regards,

Nan Jiang

Academic Editor

PLOS One

Journal Requirements:

Reviewers' comments:

Reviewer's Responses to Questions

**Comments to the Author**

Reviewer #1: (No Response)

2. Is the manuscript technically sound, and do the data support the conclusions?

Reviewer #1: Yes

3. Has the statistical analysis been performed appropriately and rigorously?

Reviewer #1: Yes

4. Have the authors made all data underlying the findings in their manuscript fully available?

Reviewer #1: Yes

5. Is the manuscript presented in an intelligible fashion and written in standard English?

Reviewer #1: Yes

Reviewer #1: The authors have made genuine revisions and adopted much of your recommended language verbatim. However, there is internal inconsistency between the revised Discussion and Conclusion, and a few legacy sentences still drift toward equivalence/sufficiency inference.

1. Discussion — Implied confirmation of “no effect”

Original sentence (lines 265–266)

“We used a higher volume, which may further strengthen that the high-volume PENG block did not enhance anesthetic or analgesic effects.”

“Further strengthen” implies confirmatory evidence of no effect, which is not justified in a superiority trial with limited power. This language subtly contradicts the carefully corrected type II error and non-equivalence statements later in the Discussion and Conclusion.

Therefore, I suggest to replace with “We used a higher volume; however, within the constraints of this superiority trial, no statistically significant enhancement of anesthetic or analgesic effects was detected.”

2. Discussion — Overinterpretation of anatomical/mechanistic explanation

Original sentence (lines 261–262)

“We have not found that the standard PENG block can block the peripheral branches of lumbar plexus even with high volume.”

Reads as a general negative claim, rather than a study-limited observation. Risks being interpreted as a mechanistic conclusion rather than a contextual explanation.

Therefore, I suggest to replace with “In the present study, we did not observe evidence suggesting that increasing volume resulted in consistent blockade of peripheral branches of the lumbar plexus.”

3. Discussion — Trend language bordering on inference

Original sentence (lines 283–284)

“Nevertheless, the conventional volume group exhibited a trend towards a delayed onset of first opioid use and a higher proportion of patients who remained opioid-free.”

“Exhibited a trend” can be read as directional inference, despite non-significance and lack of power. PLOS ONE generally prefers neutral phrasing when results are exploratory.

Therefore, I suggest to replace with “Nevertheless, numerically longer times to first opioid use and a higher proportion of opioid-free patients were observed in the conventional-volume group, although these differences were not statistically significant.”

4. Introduction — Strength of hypothesis framing (minor but relevant)

Original sentence (lines 105–106)

“We hypothesized that the dynamic pain score at 6 h after THA would be significantly lower in patients receiving the high-volume PENG block…”

While acceptable, the rest of the paper now strongly emphasises uncertainty, and this sentence slightly contrasts with the careful framing elsewhere. Not essential to change, but optional for consistency.

Therefore, it can be softened as “We hypothesized that the dynamic pain score at 6 h after THA might be lower in patients receiving the high-volume PENG block…”

**Do you want your identity to be public for this peer review?** For information about this choice, including consent withdrawal, please see our For information about this choice, including consent withdrawal, please see our Privacy Policy .

Reviewer #1: **Yes:** Dr Shah-Jalal SarkerDr Shah-Jalal Sarker

---

## [Author Response · Author response to Decision Letter 4]

27 Jan 2026

Manuscript No.: PONE-D-25-27762R3

Title: High-versus conventional-volume pericapsular nerve group (PENG) block for total hip arthroplasty: A randomized, controlled trial

Dear Dr Nan Jiang,

Thank you very much for your letter and helpful advice. We have revised the manuscript and would like to re-submit it for your consideration. We would like to express our sincere thanks to you and the reviewers for your constructive and positive comments which help us promote our manuscript a lot.

We have addressed all the comments raised by the editor and reviewers, and the amendments are highlighted in red in the revised manuscript. We have listed a point-by-point response to the editor and reviewers' comments as follows.

Your consideration of our revised manuscript for publication in your journal will be highly appreciated!

Best regards,

Zhao-Hui Chen,

Department of Anesthesiology,

Third Affiliated Hospital of Chongqing Medical University ,

No.1 Shuanghu Branch Road,

Chongqing, 401120, China;

Email: 650688@cqmu.edu.cn

Response to the reviewer

Manuscript No.: PONE-D-25-27762R3

Title: High-versus conventional-volume pericapsular nerve group (PENG) block for total hip arthroplasty: A randomized, controlled trial

Comment 1:

Discussion — Implied confirmation of “no effect”

Original sentence (lines 265–266)

“We used a higher volume, which may further strengthen that the high-volume PENG block did not enhance anesthetic or analgesic effects.”

“Further strengthen” implies confirmatory evidence of no effect, which is not justified in a superiority trial with limited power. This language subtly contradicts the carefully corrected type II error and non-equivalence statements later in the Discussion and Conclusion.

Therefore, I suggest to replace with “We used a higher volume; however, within the constraints of this superiority trial, no statistically significant enhancement of anesthetic or analgesic effects was detected.”

Response: Thank you for your helpful and valuable comment. We have revised the sentence（line267-270）. Revision details were shown below:

We used a higher volume, which may further strengthen that the high-volume PENG block did not enhance anesthetic or analgesic effects. We used a higher volume; however, within the constraints of this superiority trial, no statistically significant enhancement of anesthetic or analgesic effects was detected.

Comment 2:

Discussion — Overinterpretation of anatomical/mechanistic explanation

Original sentence (lines 261–262)

“We have not found that the standard PENG block can block the peripheral branches of lumbar plexus even with high volume.”

Reads as a general negative claim, rather than a study-limited observation. Risks being interpreted as a mechanistic conclusion rather than a contextual explanation.

Therefore, I suggest to replace with “In the present study, we did not observe evidence suggesting that increasing volume resulted in consistent blockade of peripheral branches of the lumbar plexus.”

Response: Thank you very much for your helpful and valuable comment. We have revised the description (line261-264), revision details were shown below:

We have not found that the standard PENG block can block the peripheral branches of lumbar plexus even with high volume. In the present study, we did not observe evidence suggesting that increasing volume resulted in consistent blockade of peripheral branches of the lumbar plexus.

Comment 3:

Discussion — Trend language bordering on inference

Original sentence (lines 283–284)

“Nevertheless, the conventional volume group exhibited a trend towards a delayed onset of first opioid use and a higher proportion of patients who remained opioid-free.”

“Exhibited a trend” can be read as directional inference, despite non-significance and lack of power. PLOS ONE generally prefers neutral phrasing when results are exploratory.

Therefore, I suggest to replace with “Nevertheless, numerically longer times to first opioid use and a higher proportion of opioid-free patients were observed in the conventional-volume group, although these differences were not statistically significant.”

Response: Thank you for your helpful and valuable comment. We have revised the description. (line285-290), revision details were shown below:

There was no statistically significant difference between the two groups concerning the time to first opioid use or the proportion of patients who did not require opioids within 48 hours post-operation. Nevertheless, the conventional volume group exhibited a trend towards a delayed onset of first opioid use and a higher proportion of patients who remained opioid-free. Nevertheless, numerically longer times to first opioid use and a higher proportion of opioid-free patients were observed in the conventional-volume group, although these differences were not statistically significant.

Comment 4:

Introduction — Strength of hypothesis framing (minor but relevant)

Original sentence (lines 105–106)

“We hypothesized that the dynamic pain score at 6 h after THA would be significantly lower in patients receiving the high-volume PENG block…”

While acceptable, the rest of the paper now strongly emphasises uncertainty, and this sentence slightly contrasts with the careful framing elsewhere. Not essential to change, but optional for consistency.

Therefore, it can be softened as “We hypothesized that the dynamic pain score at 6 h after THA might be lower in patients receiving the high-volume PENG block…”

Response: Thank you for your helpful and valuable comment. We revised the description (Line105-107), revision details were shown below:

We hypothesized that the dynamic pain score at 6 h after THA would be significantly lower might be lower in patients receiving the high-volume PENG block compared to those receiving the conventional-volume PENG block.

---

## [Decision Letter · Decision Letter 4]

9 Feb 2026

High-versus conventional-volume pericapsular nerve group (PENG) block for total hip arthroplasty: A randomized, controlled trial

PONE-D-25-27762R4

Dear Dr. Chen,

We’re pleased to inform you that your manuscript has been judged scientifically suitable for publication and will be formally accepted for publication once it meets all outstanding technical requirements.

Kind regards,

Nan Jiang

Academic Editor

PLOS One

Additional Editor Comments (optional):

Reviewers' comments:

Reviewer's Responses to Questions

**Comments to the Author**

Reviewer #1: All comments have been addressed

2. Is the manuscript technically sound, and do the data support the conclusions?

Reviewer #1: Yes

3. Has the statistical analysis been performed appropriately and rigorously?

Reviewer #1: Yes

4. Have the authors made all data underlying the findings in their manuscript fully available?

Reviewer #1: Yes

5. Is the manuscript presented in an intelligible fashion and written in standard English?

Reviewer #1: Yes

Reviewer #1: The authors have addressed all previous reviewer comments carefully and appropriately. Interpretive language has been revised to avoid over-inference, mechanistic claims are now clearly study-limited, and the distinction between lack of statistical significance and equivalence is explicitly acknowledged. The statistical methods, reporting, and limitations are transparent and appropriate for a superiority trial with a negative result.

I have no further methodological, statistical, or epidemiological concerns.

**Do you want your identity to be public for this peer review?** For information about this choice, including consent withdrawal, please see our For information about this choice, including consent withdrawal, please see our Privacy Policy .

Reviewer #1: **Yes:** Dr Shah-Jalal SarkerDr Shah-Jalal Sarker

---

## [Editor Report · Acceptance letter]

PONE-D-25-27762R4

PLOS One

Dear Dr. Chen,

I'm pleased to inform you that your manuscript has been deemed suitable for publication in PLOS One. Congratulations! Your manuscript is now being handed over to our production team.

Kind regards,

on behalf of

Dr. Nan Jiang

Academic Editor

PLOS One